# Treatment Sequencing and Outcome of Chronic Lymphocytic Leukemia Patients Treated at Fondazione Policlinico Universitario Agostino Gemelli IRCCS: A Thirty-Year Single-Center Experience

**DOI:** 10.3390/cancers15235592

**Published:** 2023-11-26

**Authors:** Idanna Innocenti, Alberto Fresa, Annamaria Tomasso, Michela Tarnani, Laura De Padua, Giulia Benintende, Raffaella Pasquale, Eugenio Galli, Francesca Morelli, Diana Giannarelli, Francesco Autore, Luca Laurenti

**Affiliations:** 1Department of Diagnostic Imaging, Radiation Oncology and Hematology, Fondazione Policlinico Universitario Agostino Gemelli IRCCS, 00168 Rome, Italy; idanna.innocenti@policlinicogemelli.it (I.I.); alberto.fresa@guest.policlinicogemelli.it (A.F.); eugenio.galli@guest.policlinicogemelli.it (E.G.); francesco_autore@yahoo.it (F.A.); 2Section of Hematology, Department of Radiological and Hematological Sciences, Catholic University of Sacred Heart, 00168 Rome, Italy; annamariatomasso2@gmail.com; 3Hematology Department, Ospedale Belcolle, 01100 Viterbo, Italy; michelatarnani@yahoo.it; 4Hematology Department, Fabrizio Spaziani Hospital, 03100 Frosinone, Italy; laura_dp_81@libero.it; 5Department of Medicine 5, Haematology and Oncology, Friedrich-Alexander-Universität Erlangen-Nürnberg (FAU), Universitätsklinikum, 91054 Erlangen, Germany; giuliabenintende97@gmail.com; 6Azienda Sanitaria Universitaria Friuli Centrale (ASU FC), SOC Clinica Ematologia, 33100 Udine, Italy; raffaella.pasquale@asufc.sanita.fvg.it; 7Department of Hematology, Universitá degli Studi di Firenze, 50121 Florence, Italy; francesca.morelli@yahoo.com; 8Facility of Epidemiology and Biostatistics, GSTeP, Fondazione Policlinico Universitario Agostino Gemelli IRCCS, 00168 Rome, Italy; diana.giannarelli@policlinicogemelli.it

**Keywords:** chronic lymphocytic leukemia, targeted therapy, personalized medicine

## Abstract

**Simple Summary:**

This study evaluates the impact of treatment sequencing on the outcomes of patients with chronic lymphocytic leukemia. By analyzing data from patients treated at the Fondazione Policlinico Universitario Agostino Gemelli IRCCS in Rome, we showed how the sequence of therapies has evolved over the years in first-line and relapsed/refractory patients and how this change in the treatment paradigm has affected survival. In addition, the poor outcome of patients exposed to cBTKi and BCL2i raises the question of what the best approach might be for treating high-risk patients.

**Abstract:**

Background: This monocentric retrospective study describes the treatment patterns and outcomes of chronic lymphocytic leukemia (CLL) patients. Methods: Adult CLL patients treated between 1992 and 2022 were included. The time to next treatment (TTNT) was defined as the time from the treatment’s start to the start of a subsequent therapy or death. The time to next treatment failure or death (TTNTF) was defined as the time from treatment discontinuation to the discontinuation of a subsequent therapy or death. Results: Of 637 registered patients, 318 (49.9%) received treatment. We evaluated 157 cBTKi-exposed, 34 BCL2i-exposed cBTKi-naïve, and 26 double-exposed patients. The five-year TTNT values in the cBTKi-exposed patients were 80% (median NR), 40% (median 40 months), and 21% (median 24 months) months in the first line (1L), second line (2L), and beyond the second line (>2L), respectively (*p* < 0.0001). The five-year TTNT values in the BCL2i-exposed patients were 83% (median NR), 72% (median NR), 12% (median 28 months) in the 1L, 2L, and >2L, respectively (*p* = 0.185). The median TTNTF was 9 months (range 1–87) after cBTKi and 17 months (range 8–49) after both a cBTKi and BCL2i. Conclusions: This study suggests that, in CLL patients, the earlier we used targeted therapies, the better was the outcome obtained. Nonetheless, the poor outcomes in the advanced lines of therapy highlight the need for more effective treatments.

## 1. Introduction

Chronic lymphocytic leukemia, including small lymphocytic lymphoma (CLL/SLL), is the most common leukemia in adults, with an incidence of 5 per 100,000 people/year, progressively increasing with age [1]. At the time of diagnosis, the median age of patients is 70 years and an estimated 95% present at least one medical comorbidity. Most CLL patients, approximately 70% to 80%, are asymptomatic at the time of diagnosis, and one-third will never require anti-CLL treatment [2]. The choice and sequencing of CLL treatment depend on the clinical and molecular features of the disease but also on the patient (age, performance status, comorbidities, polypharmacy, presence of a caregiver, patient preferences), previous treatments, tolerability, and adverse event profiles. Despite therapeutic advances over the past decade, CLL is not curable except with allogeneic hematopoietic cell transplantation, and repeated treatments are often necessary. In recent years, the clinical course of this disease has progressively improved with the introduction of specific targeted agents, Bruton’s tyrosine kinase inhibitor (BTKi), phosphatidylinositol 3-kinase inhibitor (PI3Ki), and B-cell lymphoma-2 inhibitor (BCL2i) +/− anti-CD20, first for patients with relapsed/refractory CLL [3,4,5,6,7,8,9,10], and later also for first-line treatment [11,12,13,14,15,16,17,18,19,20]. The covalent BTKi (cBTKi) are oral irreversible inhibitors, used indefinitely or until toxicity or disease progression. Ibrutinib, the first-in-class of cBTKi, was approved for the treatment of CLL in Italy in 2016, followed by next-generation cBTK inhibitors designed to minimize off-target binding and limit associated side effects, namely acalabrutinib, approved in 2022, and zanubrutinib, currently available for compassionate use. Phosphatidylinositol 3-kinase inhibitor idelalisib has also been approved in Italy for use in relapsed/refractory CLL [21], but its use is now limited due to the challenges associated with the management of its adverse events. Venetoclax-based treatment in association with the anti-CD20 monoclonal antibodies rituximab [10] or Obinutuzumab [19] has enabled the development of fixed-duration biologic therapeutic regimens for both relapse/refractory and first-line treatments, respectively. In subsequent years, all these drugs have been progressively incorporated into CLL therapeutic strategies, increasingly anticipating the line of therapy in which they are employed and adding other cBTKi to the roster of options. cBTKis and BCL-2i +/− anti-CD20 have transformed the therapeutic landscape for CLL/SLL, including in high-risk patients, by demonstrating prolonged PFS and OS over chemoimmunotherapy (CIT), the traditional standard of care. Therefore, whereas in the past CIT regimens were the most widely used treatments, especially in first-line treatment, currently, most patients with CLL receive completely chemo-free treatment. Nevertheless, although most patients achieve remission and do not require further therapy for years, a proportion discontinue treatment due to toxicity or progression, and, after cBTKi and cBCL2i therapies, there is no standardized therapy. To optimize personalized therapy, an emerging need is to identify patients already receiving cBTKi and BCL2i therapy, evaluate their outcome, and define what is the best long-term therapeutic strategy for them. This monocentric retrospective observational study describes the real-world data regarding the therapeutic patterns and outcomes of patients with CLL/SLL treated at the Fondazione Policlinico Universitario Agostino Gemelli IRCCS in the last 30 years, to better understand the impact of treatment sequencing in the novel targeted-agents era.

## 2. Materials and Methods

In this study, we enrolled adult CLL/SLL patients treated between 1992 and 2022, and they were prospectively registered in our institution’s electronic database. The study was carried out according to the Helsinki Declaration, Good Clinical Practice, and applicable national regulations and approved by the local Ethical Committee. All the patients provided written informed consent. The CLL patients were diagnosed and treated according to the international workshop on CLL (iwCLL) criteria.

For each patient, we collected data about the biological features of the disease (cytogenetic aberrations and molecular mutations, immunoglobulin heavy chain [IGHV] gene mutational status), the type, number, and sequence of the treatment regiments, the drug administration dates, and the outcomes. The main treatment regiments used according to international guidelines, both in treatment-naive and refractory/relapsed patients, were as follows: chlorambucil +/− steroids, cyclophosphamide +/− steroids +/− rituximab, fludarabine and cyclophosphamide +/− rituximab (FC or FCR), rituximab + cyclophosphamide + doxorubicin + vincristine + prednisone (R-CHOP), alemtuzumab +/− fludarabine, bendamustine and rituximab (BR), chlorambucil + anti-CD20 (rituximab or obinutuzumab), ibrutinib, idelalisib-rituximab, acalabrutinib, venetoclax +/− anti-CD20 (obinutuzumab, VO or rituximab, VR).

The clinical outcomes included the time to first treatment (TTFT), the time to next treatment (TTNT), and the time to next treatment failure or death (TTNTF). The TTFT was defined as the time from the diagnosis to the start of the first line of therapy. The TTNT was defined as the time from the start of the line of therapy to the start of a subsequent therapy or death or censored at the date of the last follow-up. The TTNTF was defined as the time from the discontinuation of the line of therapy to the discontinuation of a subsequent therapy or death or censored at the date of the last follow-up. The time-to-event outcomes were evaluated using the Kaplan–Meier method. The median survival time and their 95% confidence intervals (95% CI) are reported.

## 3. Results

Of 637 registered CLL patients between 1992 and 2022, 318 (49.9%) received first-line treatment, and 133 (20.9%) patients were treated for relapse/refractory. The patients’ characteristics and their exposure to the different drugs are reported in Table 1.

The median age at frontline treatment was 65 years; out of 255 tested patients, 137 had unmutated IGHV genes; out of 279 tested patients, 28 had a 17p deletion; out of 232 tested patients, 42 had a TP53 mutation, and 53 out of 227 tested patients had TP53/del 17p. The median TTFT was 24 months (range 7–51 months). When evaluating the impact of known adverse prognostic factors on the TTFT, we found that an unmutated IGHV (33 vs. 108 months, *p* < 0.001), a positive CD49d (34 vs. 89 months, *p* < 0.001), and TP53 WT (26 vs. 47 months, *p* < 0.001) were factors associated with a shorter TTFT. We stratified all the patients according to the CLL International Prognostic Index (CLL-IPI), detecting a median TTFT of 149 months, 47 months, 26 months, and 12 months for the low-, intermediate-, high-, and very high-risk patients, respectively (*p* < 0.001) (Appendix A).

The median follow-up for the treated patients was 97 months (IQR 53–168). The median observation time from first-line treatment was 52 months (IQR 22–121). When evaluating the impact of known adverse prognostic factors on the OS, we observed no association of IGHV and TP53 mutational status with a worse OS (IGHV mutated vs. unmutated median OS, 297 months vs. not reached, *p* = 0.052; TP53 disrupted vs. WT, 297 vs. 261 months, *p* = 0.18). After stratifying all the patients according to the CLL-IPI, the 8-year OS was 96.8%, 90.9%, 81.6%, and 37.9% for the low-, intermediate-, high-, and very high-risk patients, respectively (*p* < 0.001) (Appendix A).

### 3.1. Treated Cohort

The patients treated with first-line chemotherapy (CT) were fifty-three; one-hundred and sixty two patients were treated with CT + antiCD20; seventy-five were treated with cBTKi +/− anti-CD20; fourteen were treated with BCL2i +/− anti-CD20; and two patients received a cBTKi+BCL2i frontline treatment, while twelve received other treatments.

Of the 318 treated patients, in the frontline setting, 136 were IGHV unmutated: of these, 37/136 were treated with cBTKi +/− anti-CD20, 8/136 with BCL2i +/− anti-CD20, and 2/136 with cBTKi+BCL2i. In the frontline setting, 54/318 patients harbored TP53 mutations and/or del17p: of these, 22/54 were treated with cBTKi +/− anti-CD20 and 1/54 with BCL2i + anti-CD20.

The patients who received only CT or CT + antiCD20 had a shorter TTNT than the cBTKi- and BCL2i-treated groups. After 3 years, the TTNT was 47.1%, 69.3%, 89.6%, and 83.3% for CT, CT+antiCD20, cBTKi, and BCL2i, respectively, with a median TTNT of 33 and 58 months for CT and CT+antiCD20, while it was not reached for the targeted agents (*p* = 0.006) (Figure 1A). As expected, CT showed a shorter TTNT in all the subgroups, while, instead, CIT resulted in worse outcomes in the subgroups of patients with unmutated IGHV and disrupted del17p/TP53. The three-year TTNT in the unmutated IGHV patients was 18.8%, 60.7%, 90.8%, and 100% for CT, CT+antiCD20, cBTKi, and BCL2i, respectively (*p* < 0.0001) (Figure 1B). The two-year TTNT in the disrupted del17p/TP53 patients was 75.0%, 52.6%, 93.3%, and 100% for CT, CT+antiCD20, cBTKi, and BCL2i, respectively (*p* < 0.0001) (Figure 1C). We then evaluated the impact of CLL-IPI at the moment of treatment on the TTNT. The median TTNT was not affected by a higher CLL-IPI in the patients treated with inhibitors, while it was significantly associated with worse survival in the patients treated with CT +/− antiCD20 (Table 2, Figure 2). CD49d expression was not associated with a worse TTNT, independently from the therapeutic strategy (median TTNT 57.2 months (95% CI: 43.7–71.3) vs. 64.2 months (95% CI: 57.5–70.9) for CD49d-negative and CD49d-positive patients, respectively, *p* = 0.38).

In 133 relapsed/refractory patients, different strategies were adopted depending on the year of the treatment and the previous line of treatment.

After a first-line therapy with CT+antiCD20, the patients receiving cBTKi or BCL2i +/− antiCD20 therapy showed a longer TTNT compared to the patients treated with CT +/− antiCD20, especially in the unmutated IGHV patients and regardless of their del17p/TP53 mutational status (Figure 3). The median TTNT at second-line treatment was 93.5 (IQR 68.3–118.7) and 67.9 (IQR 62.1–73.7) months for the targeted agents and CT +/− antiCD20, respectively (*p* = 0.019). The IGHV unmutated patients showed a second-line TTNT of 90.6 (IQR 67.8–113.4) and 62.6 (54.9–70.3) months for the targeted agents and CT +/− antiCD20, respectively (*p* = 0.047). The patients without del(17p)/TP53 disruptions showed a second-line TTNT of 109.7 (IQR 85.1–134.3) and 62.6 (IQR 59.6–65.6) months for the targeted agents and CT +/− antiCD20, respectively (*p* < 0.0001). Only 16/53 patients with a del(17p)/TP53 disruption received second-line therapy, so the analysis for this subgroup did not achieve sufficient statistical power to be significant.

The treatment sequence of the entire cohort is shown in the Sankey plot in Figure 4, while Table 3 shows the different types of treatment prescribed to the treatment-naïve and relapsed/refractory patients, stratified by year. The toxicities in the different treatment cohorts are listed in Appendix A. We detected a difference between cBTKi and BCL2i only in hematological toxicities in the second line of treatment, whereas no differences in extra-hematological adverse events were reported.

### 3.2. cBTKi- and/or BCL2i-Exposed

The characteristics of the treated cohorts, divided on the basis of the drug received and the line of therapy, are listed in Appendix A.

The five-year TTNT values in the cBTKi-exposed patients were 80% (median not reached), 40% (median 40 months), and 21% (median 24 months) in the first, second, and subsequent lines of treatment, respectively (*p* < 0.0001). The five-year TTNT values in the BCL2i-exposed patients were 83% (median not reached), 72% (median not reached), and 12% (median 28 months) in the first, second, and subsequent lines of treatment, respectively (*p* = 0.185). By stratifying the BCL2i-exposed patients in the first line of treatment and those who were relapsed/refractory, the 5-year TTNT values were 83% (median not reached) and 42.5% (median 30 months) (*p* = 0.089). The Kaplan–Meier curves are shown in Figure 5.

When analyzing the patients who progressed after the targeted therapies, the median TTNTF was 9 months (range 1–87) for all the patients who had discontinued the cBTKi, independently of the line of treatment, and 17 months (range 8–49) for those who had discontinued both a cBTKi and a BCL2i. To note, four patients in this group underwent allogeneic hematopoietic stem cell transplantation (HSCT) for high-risk CLL features or Richter’s transformation.

## 4. Discussion

Over the years and with different treatment regimens, the factors that drive the choice of treatment in treatment-naïve and R/R CLL patients are changing. In the era of CT and CIT, biological characteristics, age, performance status, and comorbidities are important in the treatment choice and in the outcomes. In the era of target therapy, the biological features of the disease and therapeutic sequencing are the principal factors guiding the treatment choice and the outcomes. Optimizing the treatment sequence in CLL and evaluating the outcome of patients who have received cBTKi and BCL2i are critical to improving treatment strategies. Since not all the patients who progress need retreatment for CLL if they do not meet the iwCLL treatment criteria [22], we chose to assess outcome on the need to be retreated rather than on disease progression in order to explore the true need for newer treatment options of the population treated with cBKTi and BCL2i.

In this study, data from patients treated in our Center demonstrate how the anticipation of targeted therapy improves its performance, as shown by the significantly better first-line TTNTs for cBTKis. This consideration is stronger for cBTKis, for which we have a longer observation time, and needs to be confirmed for BCL2i, which has received more recent approval in Italy. As a result, in our cohort, frontline BCL2i seems to have outperformed in the IGHV-unmutated and/or TP53-disrupted patients compared to the trials, probably due to the shorter observation time (median OT 10 months). Nevertheless, overall, the results showed the superiority of targeted agents over CT and CIT, especially in IGHV-unmutated and del(17p)/TP53-disrupted patients. For the patients with a “low-risk” disease (i.e., IGHV mutated, TP53 WT, and del17p absent), inhibitor therapy did not show a clear superiority over CIT in our cohort. In the trial results, however, PFS superiority was also shown in these patients, with both ibrutinib in RESONATE-2 (HR 0.174; 95% CI, 0.089 to 0.342) [13] and venetoclax-obinutuzumab in CLL14 in unfit patients versus Chlorambucil-obinutuzumab (HR 0.36; 95% CI, 0.19 to 0.68) [23]. In contrast, neither venetoclax-obinutuzumab therapy in CLL13 in fit patients versus FCR was superior [24], nor zanubrutinib versus BR in SEQUOIA [18], nor continued therapy with acalabrutinib versus chlorambucil-obinutuzumab in ELEVATE-TN [15]. Therefore, it remains to be discussed individually, especially for young patients with a low-risk disease, whether to perform CIT or fixed-duration therapy with inhibitors.

CLL-IPI is one of the most effective scores for predicting TTFT and OS in newly diagnosed CLL [25,26]. In our cohort, we confirmed its prognostic value at diagnosis. Moreover, we evaluated the impact of the CLL IPI calculated at the time of treatment on the TTNT, stratifying patients according to the type of treatment received. Previous real-life experiences demonstrated the prognostic value of CLL IPI on PFS in patients treated with CIT [27], but no validation on a population treated with novel drugs. We confirmed the impact on survival in the patients treated with CIT, but we also showed how this stratification is no longer prognostic in patients treated with inhibitors. Our hypothesis is that mitigating the prognostic impact of the high-risk molecular features of CLL nullifies the predictive outcome of CLL-IPI.

Focusing on targeted agents, the 2-year PFS for first-line treatment across different trials was about 88% for Ibrutinib [11,12,14], 87% for acalabrutinib [16], 85% for zanubrutinib [18], and 88% for venetoclax-obinutuzumab [19]. In the relapsed/refractory setting, for Ibrutinib, the 5-year PFS was 40% [3]; for acalabrutinib, the 42-months PFS was 62% [4]; for zanubrutinib, the 2-year PFS was 78.4% [6], and, for venetoclax rituximab, the 5-year PFS was 37.8% [10]. In our cohort, the 5-year TTNT after frontline cBTKi was 80%, while, after BCL2i, it was 83%; in the relapsed/refractory patients, the 5-year TTNT after cBTKi was 32.4%, while, after BCL2i, it was 42.5%. Thus, the results of our cohort are aligned with the different trials, also in TP53-disrupted patients [28,29], although caution should be used when comparing the results, both because of the time-dependent variable chosen and the nature of the study itself. In addition, these data point to the growing need to optimize the sequence of these drugs, especially considering the ever-closer possibility of combining cBTKi and BCL2i in the first-line setting with or without antiCD20 based on the promising results of several ongoing trials [24,30,31,32].

By stratifying the different types of treatment by year, we note how CIT has progressively declined in favor of targeted agents. This consideration is most noticeable in the relapsed/refractory patients, of whom only a fraction has not performed targeted therapy in the past 3 years. Given the recent approval in Italy of several first-line targeted treatments, it is likely that we will notice this progressive reduction of the role of CIT in this setting as well in the coming years. It is also interesting to note that the median age at treatment is progressively increasing over the years, probably due to the greater tolerability of targeted therapies which allow the treating of older patients.

Despite the excellent efficacy shown by currently available treatments, there remains a proportion of patients for whom they are not sufficient. Data from our cohort confirm that, already after the first targeted therapy, the subsequent therapeutic chances prove to be less effective (median TTNTF after cBTKi 9 months). Our results were consistent with previous experiences for patients relapsing after cBTKi, considering that our cohort was mostly composed of patients taking cBTKi in the first or second line of treatment [33,34]. Previously reported TTNTF in double-exposed patients was inferior (5.5 months) [33] compared to our cohort (median TTNTF 17 months) probably for two main reasons: firstly, four out of six patients in this group underwent allogeneic HSCT for high-risk CLL features or Richter’s transformation; secondly, patients treated with BCL2i have a shorter observation time because of its later approval in Italy. Nevertheless, these data reinforce the idea that, considering the worsening outcome after multiple targeted agents, patients in the condition to receive treatment urgently need new viable alternatives after current treatments have been exhausted.

The strengths of this study are the single-center nature of its data collection, ensuring uniformity in the diagnosis and treatment criteria of patients managed by a single team, the certainty of the lines of therapy in which targeted agents were received, and the accurately recorded dates of initiation and discontinuation. The limitations inherent in the study are mainly the limited availability of long-term data due to the delay in approval in Italy compared to the FDA, the retrospective nature of the data, and the small cohort due to the single-center nature of the data. Despite its limitations, this real-life study demonstrated how anticipating target therapy improved the outcome of CLL, including high-risk patients, and how outcomes progressively worsened as the treatment options for CLL became exhausted. In addition to the development of optimal therapeutic sequencing, the poor outcomes in advanced lines of therapy highlight the need for even more effective treatments, especially for younger and high-risk patients.

## 5. Conclusions

These results suggest that the use of targeted agents leads to better outcomes if used earlier in the treatment algorithm. The choice of the agent, especially in the first line of treatment, is influenced by the biologic characteristics of the CLL and by patients’ age and fitness status. Apart from allogeneic HSCT, which can only be reserved for a very small proportion of patients, novel therapies and combinations are under development or approval, and it is critical to integrate them into optimal treatment sequencing.

## Figures and Tables

**Figure 1 cancers-15-05592-f001:**
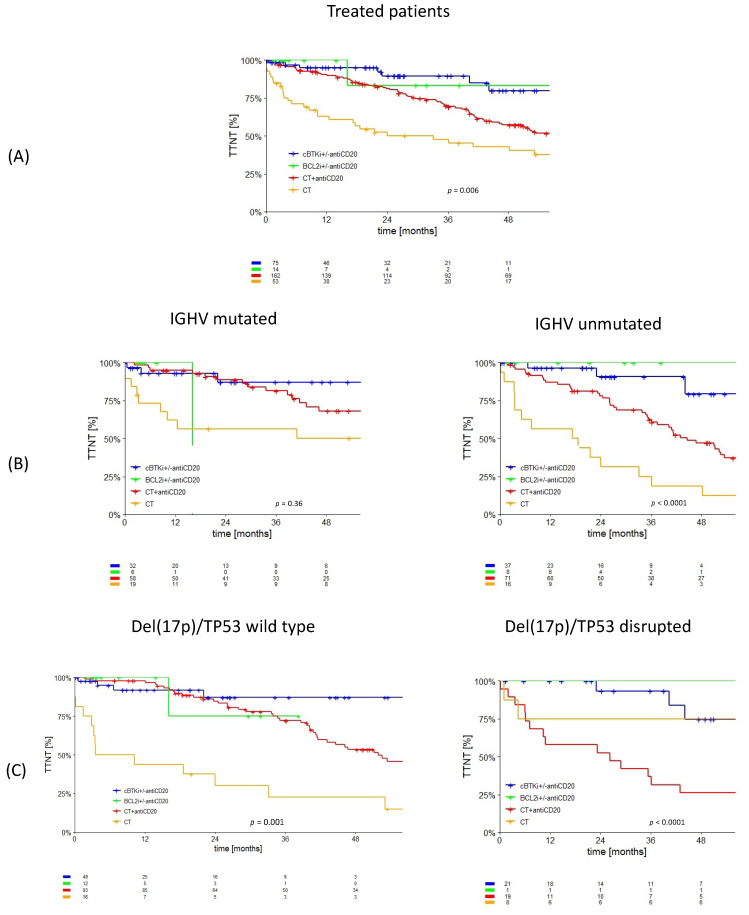
TTNT of the patients treated in first-line therapy (not considering the patients treated with BTKi+BCL2i, only two patients, or another treatment, *n* = 12). (**A**) TTNT of the entire cohort according to the treatment received. (**B**) TTNT according to the treatment received and IGHV mutational status. (**C**) TTNT according to the treatment received and del(17p)/TP53 mutational status.

**Figure 2 cancers-15-05592-f002:**
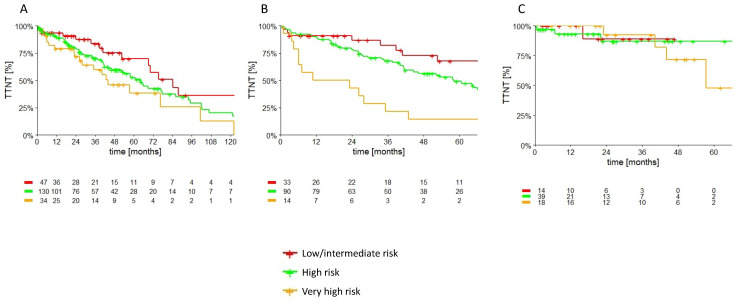
Patients’ TTNT according to the CLL-IPI calculated at treatment. (**A**) TTNT of the entire cohort. (**B**) TTNT of CT +/− anti-CD20. (**C**) TTNT of the inhibitors’ cohort.

**Figure 3 cancers-15-05592-f003:**
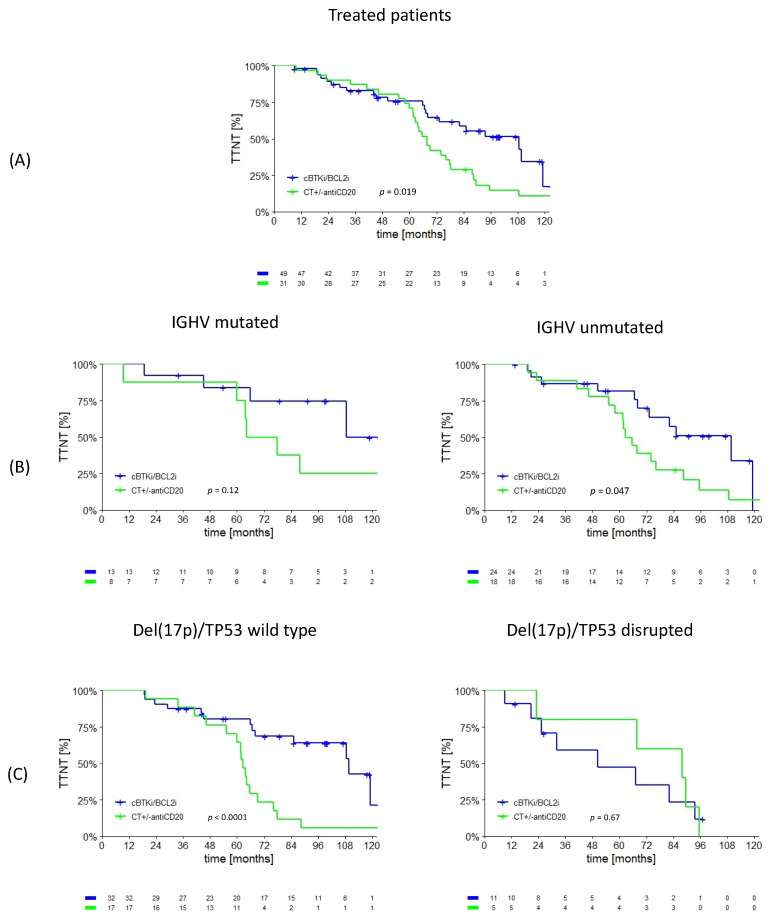
TTNT of patients treated in second-line therapy with a biological agent or CT +/− antiCD20 after frontline chemoimmunotherapy. (**A**) TTNT of the entire second-line cohort according to the treatment received. (**B**) TTNT according to the treatment received and IGHV mutational status. (**C**) TTNT according to the treatment received and del(17p)/TP53 mutational status.

**Figure 4 cancers-15-05592-f004:**
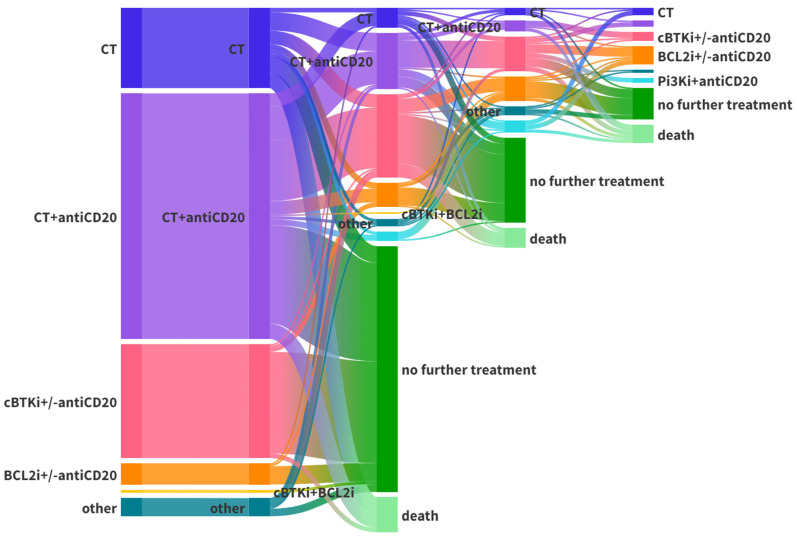
Sankey plot of the therapy of the whole cohort, before and after cBTKi/BCL2i.

**Figure 5 cancers-15-05592-f005:**
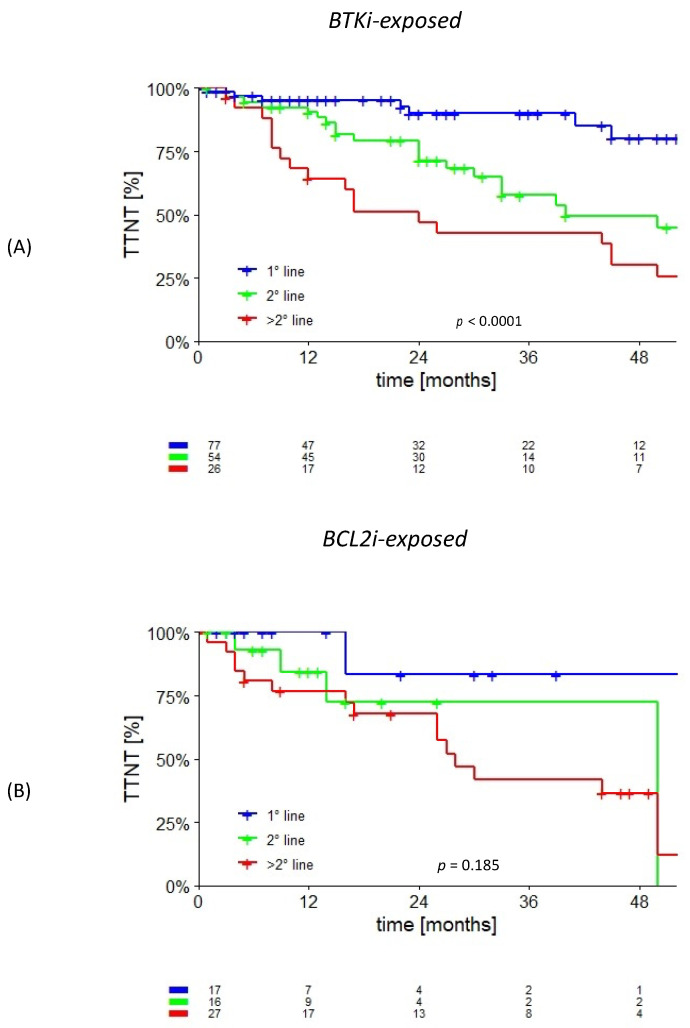
Time to next treatment in the BTKi-exposed (**A**) and BCL2i-exposed (**B**) patients according to the line of treatment.

**Table 1 cancers-15-05592-t001:** Patients’ characteristics.

		Entire Cohort (*n* = 637)	Treated Cohort (*n* = 318)
**Age, years (median, IQR)**		64 (56–72)	65 (58–72)
**Gender, *n* (%)**	Male	369 (57.9%)	201 (63.2%)
	Female	268 (42.1%)	117 (36.8%)
**FISH analysis, *n* (%)**	Deletion 17p	30/475 (6.3%)	28/279 (10.0%)
	Deletion 13q	202/470 (43.0%)	123/276 (44.6%)
	Deletion 11q	42/470 (8.9%)	36/276 (13.0%)
	Trisomy 12	72/470 (15.3%)	43/276 (15.6%)
**TP53, *n* (%)**	Mutated	46 (7.2%)	42 (13.2%)
	Unmutated	280 (44.0%)	190 (59.7%)
	Missing	311 (48.8%)	86 (27.0%)
**Del (17p) and/or TP53 mut, *n* (%)**		59 (9.3%)	53 (16.7%)
**IGHV, *n* (%)**	Mutated	246 (38.6%)	118 (37.1%)
	Unmutated	170 (26.7%)	137 (43.1%)
	Missing	221 (34.7%)	63 (19.8%)
**First-line treatment, *n* (%)**	No	319 (50.1%)	/
	Yes	318 (49.9%)	318 (100%)
**Time to Treatment, months (median, IQR)**		/	24 (7–51)
**cBTKi- and BCL2i-naïve, *n* (%)**		/	127 (39.9%)
**cBTKi-exposed, *n* (%)**		/	157 (49.4%)
**BCL2i-exposed, *n* (%)**		/	60 (18.9%)
**BCL2i-exposed cBTKi-naïve, *n* (%)**		/	34 (10.7%)
**cBTKi- and BCL2i- exposed, *n* (%)**		/	26 (8.2%)
**CIT, cBTKi-, and BCL2i-exposed, *n* (%)**		/	17 (5.3%)
**First-line Treatment, *n* (%)** *cBTKi +/*− *antiCD20* *BCL2i +/*− *antiCD20* *cBTKi+BCL2i* *CT+antiCD20* *CT* *Other*		/	*75 (23.6%)* *14 (4.4%)* *2 (0.6%)* *162 (50.9%)* *53 (16.7%)* *12 (3.8%)*

cBTKi: covalent Bruton’s tyrosine kinase inhibitors; BCL2i: B-cell lymphoma-2 inhibitor; CIT: chemoimmunotherapy; and CT: chemotherapy.

**Table 2 cancers-15-05592-t002:** Patients’ TTNT according to the CLL-IPI calculated at treatment.

	Treated Cohort	*p*	CT +/− anti-CD20 Cohort	*p*	Inhibitors Cohort	*p*
**Low/intermediate risk** **CLL-IPI at treatment** **3-year TTNT**	83.8% (95% CI: 73.4–94.2)	0.052	82.2% (95% CI: 67.7–96.7)	<0.001	88.9% (95% CI: 68.3–100.0)	0.88
**High risk** **CLL-IPI at treatment** **3-year TTNT**	70.3% (95% CI: 61.3–79.3)		67.9% (95% CI: 57.9–77.9)		87.0% (95% CI: 72.5–100.0)	
**Very high risk** **CLL-IPI at treatment** **3-year TTNT**	59.8% (95% CI: 41.8–77.8)		21.4% (95% CI: 0.0–43.0)		92.3% (95% CI: 77.8–100.0)	

**Table 3 cancers-15-05592-t003:** Prescriptive preferences and median age at treatment.

Cohort	Median Age (Years)	CT +/− antiCD20, *n* (%)	cBTKi +/− antiCD20, *n* (%)	BCL2 +/− antiCD20, *n* (%)	cBTKi+BCL2, *n* (%)	PI3Ki, *n* (%)	Other, *n* (%)	Total
	**Frontline treatment**							
**1992–2013**	63	68 (94.4)	0	0	0	0	4 (5.6)	72
**2014–2019**	69	105 (77.2)	25 (18.4)	2 (1.5)	0	0	4 (2.9)	136
**2020–2023**	70	41 (37.7)	50 (45.9)	12 (11.0)	2 (1.8)	0	4 (3.7)	109
	**Relapsed/refractory treatment**							
**1992–2013**	62	14 (70.0)	0	0	0	0	6 (30.0)	20
**2014–2019**	69	38 (30.6)	50 (40.3)	16 (12.9)	1 (0.8)	16 (12.9)	3 (2.4)	124
**2020–2023**	69	18 (20.7)	32 (36.8)	30 (34.5)	0	1 (1.1)	6 (6.9)	87

## Data Availability

Data are available upon request.

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
