# Peer review of "Treatment Sequencing and Outcome of Chronic Lymphocytic Leukemia Patients Treated at Fondazione Policlinico Universitario Agostino Gemelli IRCCS: A Thirty-Year Single-Center Experience"

_cancers, 2023, doi:10.3390/cancers15235592_

Round 1

Reviewer 1 Report

Comments and Suggestions for Authors

Innocenti and Fresa et al described a retrospective analysis of a single center institution. Authors should be acknowledged for carefully collected consecutive CLL patients during years. Overall, this study confirms what is already know about the better outcome of pathways inhibitors compared to CIT.

Few comments: 

·      In the first part of results authors may state whether the well know CLL prognostic markers such as IGHV mutational status and TP53 disruption impact time to first treatment and overall survival in the 637 registered patients. In case, survival curves should be added in supplementary material. 

·      Figure 1B. BCL2i treated patients are only 14 and the follow-up is short. The flat TTNT curve of unmutated IGHV patients treated with BCL2i needs clarification since is in contrast with results from clinical trials. 

·      Figure 1C. The legend covers a part of the survival curve of CT treated patients.

·      The authors may add in the discussion a section related to the use of CIT in IGHV mutated patients stating also the latest data from clinical trials and from real-life analysis. 

·      Have the authors enough statistical power to perform TTNT curves comparing patients who received 1stline BTKi and 2nd line BCL2i and vice versa?

·      Can the authors provide adverse events and compare them between BTKi and BCL2i treated patients? 

Comments on the Quality of English Language

None

Author Response

Dear reviewer,

Thank you for your kind suggestions and the opportunity to revise our paper on ‘Treatment sequencing and outcome of chronic lymphocytic leukemia patients treated at Fondazione Policlinico Universitario Agostino Gemelli IRCCS: a thirty-year single-center experience’. The suggestions offered have been immensely helpful.

I have included your comments immediately after this letter and responded to them individually, indicating exactly how we addressed each concern or problem and describing the changes we have made.

The revisions have been approved by all authors.

We hope the revised manuscript will better suit Cancers but are happy to consider further revisions, and we thank you for your continued interest in our research.

Sincerely,

Luca Laurenti, MD

MD, Department of Hematology

Fondazione Policlinico Universitario A. Gemelli IRCCS, Roma, Italia

Largo A. Gemelli, 8. Rome, Italy. 00168

Email: luca.laurenti@unicatt.it

Phone: +39 0630154180

Reviewer Comments, Author Responses and Manuscript Changes

Comment 1: ‘In the first part of results authors may state whether the well know CLL prognostic markers such as IGHV mutational status and TP53 disruption impact time to first treatment and overall survival in the 637 registered patients. In case, survival curves should be added in supplementary material’

Response: Thank you for the suggestion. We performed the analyses and added the results and the curves as suggested.

Comment 2: ‘Figure 1B. BCL2i treated patients are only 14 and the follow-up is short. The flat TTNT curve of unmutated IGHV patients treated with BCL2i needs clarification since is in contrast with results from clinical trials.’

Response: According to the reviewer suggestion, we added a paragraph in the discussion explaining the outcome of high-risk patients on venetoclax therapy in our cohort.

Comment 3: ‘Figure 1C. The legend covers a part of the survival curve of CT treated patients’.

Response: Thank you for the suggestion. We modified the picture as requested.

Comment 4: ‘The authors may add in the discussion a section related to the use of CIT in IGHV mutated patients stating also the latest data from clinical trials and from real-life analysis’.

Response: Thank you for the suggestion. We added a paragraph on low-risk patients in the discussion.

Comment 5: ‘Have the authors enough statistical power to perform TTNT curves comparing patients who received 1stline BTKi and 2nd line BCL2i and vice versa?’

Response: Thank you for the suggestion. Unfortunately, due to the short follow-up of frontline patients with venetoclax, we don’t have enough patients to perform the requested analysis.

Comment 6: ‘Can the authors provide adverse events and compare them between BTKi and BCL2i treated patients?’

Response: Thank you for the suggestion. We added a table and a paragraph on adverse events.

Reviewer 2 Report

Comments and Suggestions for Authors

The authors analyzed efficacy of therapeutic agents era by era as well as the results of treatment sequencing for CLL patients in a single institute. Although retrospective study, the number of CLL patients analyzed are sufficient and these patients were treated with appropriate agents in each era. The results obtained are interesting and may be useful for future strategy for CLL treatment. However, the manuscript has some problems to be corrected.

Major comments:

1.       The conclusions in the Abstract are rather vague. Please draw more definite conclusions. For example, these results suggest that cBTKi + antiCD20 and BCL2 + are equipotent at any lines, etc. Also, please suggest favorite treatment sequences in the conclusions if possible.

2.       5. Conclusions in page 10: these are not conclusive and may be moved to Dicsussion section. Pease draw more definite conclusions in relation to the Abstract conclusions.

3.       Figures (A-C) in Figure 1 are too small. The readers would not be able to read the letters. Please improve.

4.       Table 1: FISH analysis, n→FISH analysis, n (%).

Please show all the data (TP53, Del (17p), IGHV, and First line) as FISH analysis, n (%), Deletion 17p 30/475 (6.3%), 28/279 (10.0%).

Minor comments:

1.       Please don’t use capital letter at the head of word unless the word is proper noun throughout the manuscript.

2.       The following abbreviations should be first written in full term then abbreviate in the parenthesis; IGHV (page 3), TN (page 9), CIT (page 9). On the contrary, chemoimmunotherapy (page 9, line 18 from the bottom) should be written as CIT).

3.       In the last years..(page 2, line 11) →In recent years..

4.       Page 3: Is CG (chlorambucil + obinutuzumab) correct?  Is VG (venetoclax + Obinutuzumab) correct?  

steroids+/.rituximab+→steroids+/-rituximab (page 3, line 8).

5. Legends for Table 1: CIT o CT: chemoimmunotherapy→CT: chemotherapy?

Comments on the Quality of English Language

Generraly, the manuscript is appropriately written and only minor editing of English is required. 

Author Response

Dear reviewer,

Thank you for your kind suggestions and the opportunity to revise our paper on ‘Treatment sequencing and outcome of chronic lymphocytic leukemia patients treated at Fondazione Policlinico Universitario Agostino Gemelli IRCCS: a thirty-year single-center experience’. The suggestions offered have been immensely helpful.

I have included your comments immediately after this letter and responded to them individually, indicating exactly how we addressed each concern or problem and describing the changes we have made.

The revisions have been approved by all authors.

We hope the revised manuscript will better suit Cancers but are happy to consider further revisions, and we thank you for your continued interest in our research.

Sincerely,

Luca Laurenti, MD

MD, Department of Hematology

Fondazione Policlinico Universitario A. Gemelli IRCCS, Roma, Italia

Largo A. Gemelli, 8. Rome, Italy. 00168

Email: luca.laurenti@unicatt.it

Phone: +39 0630154180

Reviewer Comments, Author Responses and Manuscript Changes

Comment 1: ‘The conclusions in the Abstract are rather vague. Please draw more definite conclusions. For example, these results suggest that cBTKi + antiCD20 and BCL2 + are equipotent at any lines, etc. Also, please suggest favorite treatment sequences in the conclusions if possible. Conclusions in page 10: these are not conclusive and may be moved to Dicsussion section. Pease draw more definite conclusions in relation to the Abstract conclusions.’’

Response: Thank you for the kind suggestion. We tried to draw more precise conclusions as suggested. Nevertheless, we cannot comment precisely on the recommended sequencing given the current phase of ongoing change in the CLL treatment algorithm.

Comment 2: ‘Figures (A-C) in Figure 1 are too small. The readers would not be able to read the letters. Please improve’.

Response: Thank you for the suggestion. We improved the figure as suggested.

Comment 3: ‘Table 1: FISH analysis, n→FISH analysis, n (%). Please show all the data (TP53, Del (17p), IGHV, and First line) as FISH analysis, n (%), Deletion 17p 30/475 (6.3%), 28/279 (10.0%)’.

Response: Thank you for the suggestion. We modified the table accordingly.

Comment 4: ‘Please don’t use capital letter at the head of word unless the word is proper noun throughout the manuscript’

Response: We modified the manuscript accordingly.

Comment 5: ‘The following abbreviations should be first written in full term then abbreviate in the parenthesis; IGHV (page 3), TN (page 9), CIT (page 9). On the contrary, chemoimmunotherapy (page 9, line 18 from the bottom) should be written as CIT). In the last years..(page 2, line 11) →In recent years. Page 3: Is CG (chlorambucil + obinutuzumab) correct? Is VG (venetoclax + Obinutuzumab) correct? steroids+/.rituximab+→steroids+/-rituximab (page 3, line 8). Legends for Table 1: CIT o CT: chemoimmunotherapy→CT: chemotherapy?’

Response: Thank you for all the revisions. We corrected the manuscript as suggested.

Reviewer 3 Report

Comments and Suggestions for Authors

The authors report treatment sequencing and outcome of chronic lymphocytic leukemia patients.

1.     CD38 expression, ZAP-70 expression, and β2 -microglobulin levels are related to prognosis in chronic lymphocytic leukemia. The authors should provide the clinical data and describe it in detail.

Author Response

Thank you for your kind suggestions and the opportunity to revise our paper on ‘Treatment sequencing and outcome of chronic lymphocytic leukemia patients treated at Fondazione Policlinico Universitario Agostino Gemelli IRCCS: a thirty-year single-center experience’. The suggestions offered have been immensely helpful.

I have included your comments immediately after this letter and responded to them individually, indicating exactly how we addressed each concern or problem and describing the changes we have made.

The revisions have been approved by all authors.

We hope the revised manuscript will better suit Cancers but are happy to consider further revisions, and we thank you for your continued interest in our research.

Sincerely,

Luca Laurenti, MD

MD, Department of Hematology

Fondazione Policlinico Universitario A. Gemelli IRCCS, Roma, Italia

Largo A. Gemelli, 8. Rome, Italy. 00168

Email: luca.laurenti@unicatt.it

Phone: +39 0630154180

Reviewer Comments, Author Responses and Manuscript Changes

Comment 1: ‘CD38 expression, ZAP-70 expression, and β2-microglobulin levels are related to prognosis in chronic lymphocytic leukemia. The authors should provide the clinical data and describe it in detail.’

Response: Thank you for the suggestion. We didn’t perform the analysis on CD38 and ZAP 70 because in the era of novel drugs our laboratory does not provide that information anymore. Anyway, we analyzed the impact of CLL-IPI (therefore including β2-microglobulin levels) on TTFT, TTNT and OS.

Reviewer 4 Report

Comments and Suggestions for Authors

In this manuscript, Innocenti et al. have investigated the impact of treatment sequence on the outcome of patients with chronic lymphocytic leukemia (CLL) over a very long period, from 1992 to 2022, by covering different eras in terms of diagnosis and treatments. The manuscript is of interest and data are well-presented.

The Authors have shown the relationship between treatments and some molecular/cytogenetics factors (e.g., IGHV mutational status); however, recent studies have demonstrated that other factors, such as CD49d and CD38 positivity by flow cytometry, could also concur to prognostic definition. Therefore, is it possible to stratify patients also based on CLL-IPI, IPS-E, and ISS+CLL-IPI scores?

Author Response

Dear reviewer,

Thank you for your kind suggestions and the opportunity to revise our paper on ‘Treatment sequencing and outcome of chronic lymphocytic leukemia patients treated at Fondazione Policlinico Universitario Agostino Gemelli IRCCS: a thirty-year single-center experience’. The suggestions offered have been immensely helpful.

I have included your comments immediately after this letter and responded to them individually, indicating exactly how we addressed each concern or problem and describing the changes we have made.

The revisions have been approved by all authors.

We hope the revised manuscript will better suit Cancers but are happy to consider further revisions, and we thank you for your continued interest in our research.

Sincerely,

Luca Laurenti, MD

MD, Department of Hematology

Fondazione Policlinico Universitario A. Gemelli IRCCS, Roma, Italia

Largo A. Gemelli, 8. Rome, Italy. 00168

Email: luca.laurenti@unicatt.it

Phone: +39 0630154180

Reviewer Comments, Author Responses and Manuscript Changes

Comment 1: ‘The Authors have shown the relationship between treatments and some molecular/cytogenetics factors (e.g., IGHV mutational status); however, recent studies have demonstrated that other factors, such as CD49d and CD38 positivity by flow cytometry, could also concur to prognostic definition. Therefore, is it possible to stratify patients also based on CLL-IPI, IPS-E, and ISS+CLL-IPI scores?’

Response: Thank you for the suggestion. We didn’t perform the analysis on CD38 because in the era of novel drugs our laboratory does not provide that information anymore. Anyway, we analyzed the impact of CD49d on TTFT, TTNT and OS, and we stratified patients according to the CLL-IPI, adding both the analyses in the results.

Round 2

Reviewer 2 Report

Comments and Suggestions for Authors

 The authors have reasonably revised the previous manuscript. However, there are some issues to be corrected. Although these are very minor, but are corrected as an appropriate manuscript preparation.

Minor comments:

1.       Lines 60, 71, and 105 : Please don’t use capital letter at the head of words unless they are proper noun.

2.       Line 127: know→known in all cases.

 The authors have reasonably revised the previous manuscript. However, there are some issues to be corrected. Although these are very minor, but are corrected as an appropriate manuscript preparation.

Minor comments:

1.       Lines 60, 71, and 105 : Please don’t use capital letter at the head of words unless they are proper noun.

2.       Line 127: know→known in all cases.

 The authors have reasonably revised the previous manuscript. However, there are some issues to be corrected. Although these are very minor, but are corrected as an appropriate manuscript preparation.

Minor comments:

1.       Lines 60, 71, and 105 : Please don’t use capital letter at the head of words unless they are proper noun.

2.       Line 127: know→known in all cases.

 The authors have reasonably revised the previous manuscript. However, there are some issues to be corrected. Although these are very minor, but are corrected as an appropriate manuscript preparation.

Minor comments:

1.       Lines 60, 71, and 105 : Please don’t use capital letter at the head of words unless they are proper noun.

2.       Line 127: know→known in all cases.

 The authors have reasonably revised the previous manuscript. However, there are some issues to be corrected. Although these are very minor, but are corrected as an appropriate manuscript preparation.

Minor comments:

1.       Lines 60, 71, and 105 : Please don’t use capital letter at the head of words unless they are proper noun.

2.       Line 127: know→known in all cases.

 The authors have reasonably revised the previous manuscript. However, there are some issues to be corrected. Although these are very minor, but are corrected as an appropriate manuscript preparation.

Minor comments:

1.       Lines 60, 71, and 105 : Please don’t use capital letter at the head of words unless they are proper noun.

2.       Line 127: know→known in all cases.

 The authors have reasonably revised the previous manuscript. However, there are some issues to be corrected. Although these are very minor, but are corrected as an appropriate manuscript preparation.

Minor comments:

1.       Lines 60, 71, and 105 : Please don’t use capital letter at the head of words unless they are proper noun.

2.       Line 127: know→known in all cases.

 The authors have reasonably revised the previous manuscript. However, there are some issues to be corrected. Although these are very minor, but are corrected as an appropriate manuscript preparation.

Minor comments:

1.       Lines 60, 71, and 105 : Please don’t use capital letter at the head of words unless they are proper noun.

2.       Line 127: know→known in all cases.

 The authors have reasonably revised the previous manuscript. However, there are some issues to be corrected. Although these are very minor, but are corrected as an appropriate manuscript preparation.

Minor comments:

1.       Lines 60, 71, and 105 : Please don’t use capital letter at the head of words unless they are proper noun.

2.       Line 127: know→known in all cases.

 The authors have reasonably revised the previous manuscript. However, there are some issues to be corrected. Although these are very minor, but are corrected as an appropriate manuscript preparation.

Minor comments:

1.       Lines 60, 71, and 105 : Please don’t use capital letter at the head of words unless they are proper noun.

2.       Line 127: know→known in all cases.

 The authors have reasonably revised the previous manuscript. However, there are some issues to be corrected. Although these are very minor, but are corrected as an appropriate manuscript preparation.

Minor comments:

1.       Lines 60, 71, and 105 : Please don’t use capital letter at the head of words unless they are proper noun.

2.       Line 127: know→known in all cases.

 The authors have reasonably revised the previous manuscript. However, there are some issues to be corrected. Although these are very minor, but are corrected as an appropriate manuscript preparation.

Minor comments:

1.       Lines 60, 71, and 105 : Please don’t use capital letter at the head of words unless they are proper noun.

2.       Line 127: know→known in all cases.

 The authors have reasonably revised the previous manuscript. However, there are some issues to be corrected. Although these are very minor, but are corrected as an appropriate manuscript preparation.

Minor comments:

1.       Lines 60, 71, and 105 : Please don’t use capital letter at the head of words unless they are proper noun.

2.       Line 127: know→known in all cases.

 The authors have reasonably revised the previous manuscript. However, there are some issues to be corrected. Although these are very minor, but are corrected as an appropriate manuscript preparation.

Minor comments:

1.       Lines 60, 71, and 105 : Please don’t use capital letter at the head of words unless they are proper noun.

2.       Line 127: know→known in all cases.

 The authors have reasonably revised the previous manuscript. However, there are some issues to be corrected. Although these are very minor, but are corrected as an appropriate manuscript preparation.

Minor comments:

1.       Lines 60, 71, and 105 : Please don’t use capital letter at the head of words unless they are proper noun.

2.       Line 127: know→known in all cases.

 The authors have reasonably revised the previous manuscript. However, there are some issues to be corrected. Although these are very minor, but are corrected as an appropriate manuscript preparation.

Minor comments:

1.       Lines 60, 71, and 105 : Please don’t use capital letter at the head of words unless they are proper noun.

2.       Line 127: know→known in all cases.

 The authors have reasonably revised the previous manuscript. However, there are some issues to be corrected. Although these are very minor, but are corrected as an appropriate manuscript preparation.

Minor comments:

1.       Lines 60, 71, and 105 : Please don’t use capital letter at the head of words unless they are proper noun.

2.       Line 127: know→known in all cases.

 The authors have reasonably revised the previous manuscript. However, there are some issues to be corrected. Although these are very minor, but are corrected as an appropriate manuscript preparation.

Minor comments:

1.       Lines 60, 71, and 105 : Please don’t use capital letter at the head of words unless they are proper noun.

2.       Line 127: know→known in all cases.

 The authors have reasonably revised the previous manuscript. However, there are some issues to be corrected. Although these are very minor, but are corrected as an appropriate manuscript preparation.

Minor comments:

1.       Lines 60, 71, and 105 : Please don’t use capital letter at the head of words unless they are proper noun.

2.       Line 127: know→known in all cases.

 The authors have reasonably revised the previous manuscript. However, there are some issues to be corrected. Although these are very minor, but are corrected as an appropriate manuscript preparation.

Minor comments:

1.       Lines 60, 71, and 105 : Please don’t use capital letter at the head of words unless they are proper noun.

2.       Line 127: know→known in all cases.

 The authors have reasonably revised the previous manuscript. However, there are some issues to be corrected. Although these are very minor, but are corrected as an appropriate manuscript preparation.

Minor comments:

1.       Lines 60, 71, and 105 : Please don’t use capital letter at the head of words unless they are proper noun.

2.       Line 127: know→known in all cases.

 The authors have reasonably revised the previous manuscript. However, there are some issues to be corrected. Although these are very minor, but are corrected as an appropriate manuscript preparation.

Minor comments:

1.       Lines 60, 71, and 105 : Please don’t use capital letter at the head of words unless they are proper noun.

2.       Line 127: know→known in all cases.

 The authors have reasonably revised the previous manuscript. However, there are some issues to be corrected. Although these are very minor, but are corrected as an appropriate manuscript preparation.

Minor comments:

1.       Lines 60, 71, and 105 : Please don’t use capital letter at the head of words unless they are proper noun.

2.       Line 127: know→known in all cases.

 The authors have reasonably revised the previous manuscript. However, there are some issues to be corrected. Although these are very minor, but are corrected as an appropriate manuscript preparation.

Minor comments:

1.       Lines 60, 71, and 105 : Please don’t use capital letter at the head of words unless they are proper noun.

2.       Line 127: know→known in all cases.

 The authors have reasonably revised the previous manuscript. However, there are some issues to be corrected. Although these are very minor, but are corrected as an appropriate manuscript preparation.

Minor comments:

1.       Lines 60, 71, and 105 : Please don’t use capital letter at the head of words unless they are proper noun.

2.       Line 127: know→known in all cases.

 The authors have reasonably revised the previous manuscript. However, there are some issues to be corrected. Although these are very minor, but are corrected as an appropriate manuscript preparation.

Minor comments:

1.       Lines 60, 71, and 105 : Please don’t use capital letter at the head of words unless they are proper noun.

2.       Line 127: know→known in all cases.

 The authors have reasonably revised the previous manuscript. However, there are some issues to be corrected. Although these are very minor, but are corrected as an appropriate manuscript preparation.

Minor comments:

1.       Lines 60, 71, and 105 : Please don’t use capital letter at the head of words unless they are proper noun.

2.       Line 127: know→known in all cases.

 The authors have reasonably revised the previous manuscript. However, there are some issues to be corrected. Although these are very minor, but are corrected as an appropriate manuscript preparation.

Minor comments:

1.       Lines 60, 71, and 105 : Please don’t use capital letter at the head of words unless they are proper noun.

2.       Line 127: know→known in all cases.

 The authors have reasonably revised the previous manuscript. However, there are some issues to be corrected. Although these are very minor, but are corrected as an appropriate manuscript preparation.

Minor comments:

1.       Lines 60, 71, and 105 : Please don’t use capital letter at the head of words unless they are proper noun.

2.       Line 127: know→known in all cases.

 The authors have reasonably revised the previous manuscript. However, there are some issues to be corrected. Although these are very minor, but are corrected as an appropriate manuscript preparation.

Minor comments:

1.       Lines 60, 71, and 105 : Please don’t use capital letter at the head of words unless they are proper noun.

2.       Line 127: know→known in all cases.

Comments on the Quality of English Language

Please refer to my minor commentd.

Author Response

Dear reviewer,

thank you for your comments, we addressed both of them as suggested.

We hope the revised manuscript will better suit Cancers but are happy to consider further revisions, and we thank you for your continued interest in our research.

Sincerely,

Luca Laurenti, MD

MD, Department of Hematology

Fondazione Policlinico Universitario A. Gemelli IRCCS, Roma, Italia

Largo A. Gemelli, 8. Rome, Italy. 00168

Email: luca.laurenti@unicatt.it

Phone: +39 0630154180

Reviewer 3 Report

Comments and Suggestions for Authors

none

Author Response

Dear reviewer, 

thank you for your revision.

Sincerely,

Luca Laurenti, MD

MD, Department of Hematology

Fondazione Policlinico Universitario A. Gemelli IRCCS, Roma, Italia

Largo A. Gemelli, 8. Rome, Italy. 00168

Email: luca.laurenti@unicatt.it

Phone: +39 0630154180